# Influence of the Core Branching Density on Drug Release from Arborescent Poly(γ-benzyl L-glutamate) End-Grafted with Poly(ethylene oxide)

**Mosa Alsehli and Mario Gauthier ***

Department of Chemistry, Institute for Polymer Research, University of Waterloo, 200 University Ave. W., Waterloo, ON N2L 3G1, Canada; mhsehli@taibahu.edu.sa
* Correspondence: mario.gauthier@uwaterloo.ca

**Abstract:** Amphiphilic dendritic copolymers of arborescent poly(γ-benzyl L-glutamate) (PBG) of generations G1 and G2, grafted at their chain ends with poly(ethylene oxide) (PEO) segments (PBG-*eg*-PEO) were synthesized, characterized, and evaluated as nanocarriers for doxorubicin (DOX). The copolymers were designed with hydrophobic PBG cores having three different branching densities and were characterized by proton nuclear magnetic resonance ($^1$H NMR) spectroscopy, size exclusion chromatography (SEC), transmission electron microscopy (TEM), and atomic force microscopy (AFM). Dynamic light scattering (DLS) measurements revealed that these amphiphilic molecules behaved like unimolecular micelles without significant aggregation in aqueous media such as phosphate-buffered saline (PBS), with diameters in the 13–29 nm range depending on the generation number and the core structure. Efficient encapsulation of DOX by these unimolecular micelles was demonstrated with drug loading capacities of up to 11.2 wt%, drug loading efficiencies of up to 67%, and pH-responsive sustained drug release, as determined by UV spectroscopy. The generation number of the copolymers and the branching density of the dendritic PBG core were found to have influenced the encapsulation and release properties of the micelles. Given the tailorable characteristics, good water dispersibility, and biocompatibility of the components used to synthesize the amphiphilic arborescent copolymers, these systems should be useful as robust nanocarriers for a broad range of therapeutic and diagnostic agents.

**Keywords:** doxorubicin; arborescent poly(benzyl L-glutamate); drug delivery; sustained release

## 1. Introduction

Over the past two decades, many new drug delivery systems have been developed to enhance the aqueous solubility of drugs, increase their circulation time in the bloodstream, and thereby increase their therapeutic efficacy [1,2]. Various drug delivery systems have been developed with different designs, including polymeric micelles, metal nanoparticles, dendrimers, liposomes, and carbon nanotubes [3–7]. Among different nanocarriers used for controlled drug delivery, polymeric micelles (e.g., amphiphilic block copolymer micelles, unimolecular micelles, and cross-linked micelles) with tunable properties have shown great potential due to their excellent characteristics that brought them to the front line in the development of drug delivery systems [8,9].

A considerable amount of research has been devoted to the design of polymeric micelles of different types, sizes, morphologies, and stabilities for drug delivery applications. Among these, micelles formed by amphiphilic block copolymers were found very attractive due to their ability to solubilize hydrophobic or poorly water-soluble drugs within their core and thus enhance drug bioavailability [10,11]. While the performance of amphiphilic block copolymer micelles has been improved significantly, in many cases, they are still sensitive to solvent composition and other parameters such as concentration and temperature. For instance, once self-assembled micelles are injected into the body, they may dissociate due

to dilution, causing the rapid release of physically encapsulated drugs, which reduces the efficacy of the treatment [12]. These issues can be resolved by stabilizing polymeric micelles via crosslinking, either in the core or the corona, or by constructing unimolecular structures that are stable regardless of their environment [13–15]. Certain types of amphiphilic dendritic copolymers with covalently bonded core and shell components can behave like unimolecular micelles. In contrast to conventional amphiphilic block copolymers forming micelles by aggregation only above their critical micelle concentration (CMC), dendritic unimolecular micelles do not dissociate upon dilution and show excellent colloidal stability in aqueous media, regardless of their concentration, while their size and morphology can be controlled precisely [16,17]. Dendritic polymers, defined as macromolecules with a tree-like multi-level branched architecture, can be divided into three main classes depending on the specific characteristics of their architecture: dendrimers, hyperbranched polymers, and dendrigraft polymers [18,19]. Due to their remarkable features, such as a compact globular topology with internal cavities and the presence of functional groups useful to fulfill different tasks simultaneously, amphiphilic dendritic polymers may be useful to replace block copolymers in drug delivery applications [20]. Among the three dendritic polymer classes, dendrigraft (or arborescent) polymers provide a good compromise between essentially monodisperse dendrimers generated by a tedious synthetic process and highly polydisperse hyperbranched polymers obtained in one-step reactions. Dendrigraft polymers are typically obtained in a generation-based scheme analogous to dendrimers but with cycles of polymerization and grafting reactions. The use of polymeric chain segments rather than small molecules as building blocks leads to very rapid growth, such that high molecular weights are obtained in only a few steps for these systems while maintaining fairly narrow molecular weight distributions ($M_w/M_n < 1.1$) [21,22]. A simplified representation of the generation-based synthesis of arborescent polymers is provided in Scheme 1.

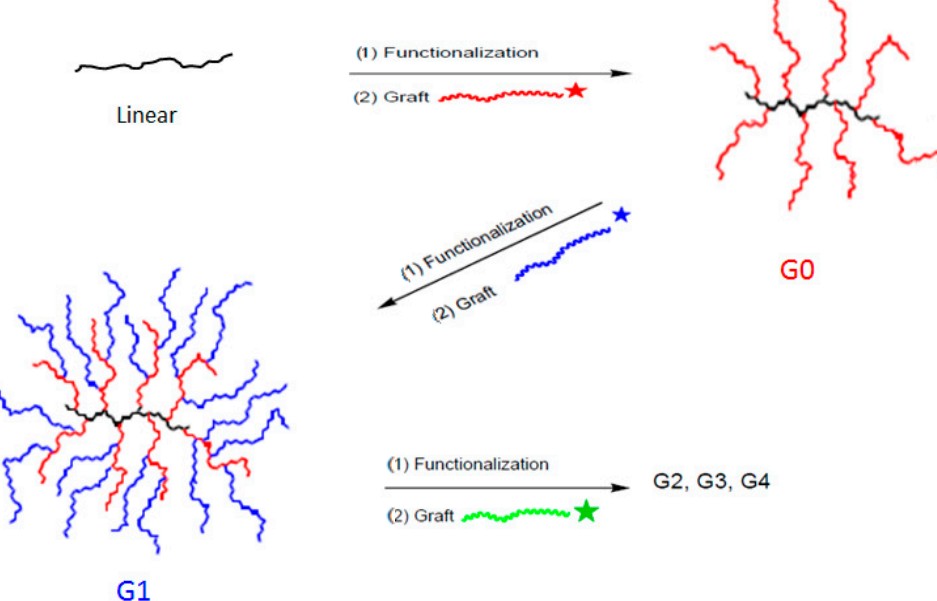

**Scheme 1.** Generation-based synthesis of arborescent polymers.

Amphiphilic arborescent copolymers can be synthesized by adding hydrophilic segments in the last grafting cycle. With proper design, these copolymers may behave like unimolecular micelles for which the size and porosity can be tailored via variations in the length of the side chains and the grafting density. Whitton and Gauthier reported the synthesis of arborescent PBG substrates grafted either randomly or at their chain ends with poly(ethylene oxide) segments to be evaluated as micellar species [23]. Unfortunately, these copolymers formed a significant amount of large aggregated species in aqueous media, so their drug encapsulation and release properties could not be explored.

This investigation focuses on the synthesis of arborescent poly($\gamma$-benzyl L-glutamate) (PBG) end-grafted with poly(ethylene oxide) (PEO) segments (PBG-*eg*-PEO) designed to avoid aggregation and their application as nanocarriers for doxorubicin (DOX) as a model hydrophobic drug. Micelles derived from PBG cores of generations G1 and G2 and three different branching densities were compared to determine the influence of these parameters on their dispersibility in aqueous media and on the encapsulation and release of the hydrophobic drug.

## 2. Materials and Methods

### 2.1. Materials

Dimethyl sulfoxide (Aldrich, Oakville, ON, Canada, ACS Reagent) and *n*-hexylamine (Aldrich, 99%) were purified by stirring overnight with $CaH_2$ (Aldrich, Reagent, 0–2 mm) and distillation under reduced pressure. *N*,*N'*-Dimethylformamide (DMF; Aldrich, >99%) serving in the polymer synthesis was purified via distillation under reduced pressure. $\gamma$-Benzyl L-glutamic acid (Bz-Glu; Bachem, Vista, CA, USA, >99%), HBr solution (Aldrich, 33% in acetic acid), *N*,*N'*-diisopropylcarbodiimide (DIC; Aldrich, 99%), $LiAlH_4$ (Aldrich, 95%), 1-hydroxybenzotriazole (HOBt; Fluka, VWR, Mississauga, ON, Canada, water content ca. 15% *w/w*), methanol (EMD Millipore, Aldrich, OmniSolv), trifluoroacetic acid (TFA, Caledon Laboratories, Georgetown, ON, Canada, 99.9%), deuterated DMSO (DMSO-$d_6$, 99.9% D, Cambridge Isotopes, Cambridge, MA, USA), deuterated water ($D_2O$; Aldrich, 99.9 atom % D), triethylamine (TEA, EMD Millipore), and doxorubicin hydrochloride (DOX·HCl, 98–102%, Aldrich) were used as received from the suppliers. Dialysis bags Spectra/Por® 7 (molecular weight cutoff MWCO 3.5 kDa) were purchased from Spectrum Laboratories Inc. (Irving, TX, USA).

### 2.2. Characterization

Proton nuclear magnetic resonance ($^1H$ NMR) spectra were obtained using a Bruker (Milton, ON, Canada) 300 MHz instrument at room temperature. $^1H$ NMR spectroscopy served to estimate the degree of polymerization of the linear PBG chains to determine the deprotection level of the PBG substrates, and to investigate the ability of the arborescent copolymers to form micelles.

Analytical size exclusion chromatography with a dual-angle light scattering detector (SEC-LS) was used for the characterization of the arborescent PBG substrates of generations G0–G2 and of the arborescent copolymers. The analytical SEC system was a Viscotek (Houston, TX, USA) GPCmax unit equipped with a VE 2001 GPC Solvent/Sample Module, a Viscotek double detector array with refractive index and dual-angle light scattering (LS) detectors, and two Agilent Technologies (Santa Clara, CA, USA) PLgel 5 $\mu$m MIXED-C and PLgel 10 $\mu$m MIXED-B organic mixed bed columns, with dimensions of 7.5 mm (ID) × 300 mm (L). The system was operated at a flow rate of 0.5 mL/min at 70 °C, using dimethyl sulfoxide (DMSO, Aldrich) with LiBr (0.05 M) as the mobile phase. Analysis of the chromatograms was performed using the OmniSEC 4.6.1 software package. Preparative SEC was carried out on a system consisting of a Waters (Milford, MA, USA) M45 HPLC pump, a 2 mL sample injection loop, a Waters R401 differential refractometer detector, and a Jordi Gel DVB 1000 Å 250 mm × 22 mm preparative SEC column (Jordi Labs, Mansfield, MA, USA). *N*,*N*-Dimethylformamide (DMF, Aldrich) with 0.2 g/L LiCl served as the mobile phase. The system was operated at a flow rate of 3 mL/min at room temperature (25 °C).

Dynamic light scattering measurements were performed using a Zetasizer Nano ZS90 (Malvern Instruments, Malvern, WR, UK) equipped with a 4 mW He–Ne laser operating at 633 nm and 25.0 °C at a scattering angle of 90°. The concentration of the solutions was around 0.5, 0.2, and 0.1 mg·mL$^{-1}$ for the G0, G1, and G2 copolymers, respectively. All the solutions were filtered with either polytetrafluoroethylene (PTFE) or cellulose acetate membrane filters having a nominal pore size of 0.45 $\mu$m prior to the measurements. The light scattering data were analyzed with the Zetasizer 7.11 Software (Malvern Instruments). Each sample was measured in triplicate.

Absorption spectra were obtained using a Cary 100 Bio UV-Vis spectrophotometer (Agilent, Santa Clara, CA, USA) with a spectral bandwidth (SBW) of 2 nm, operated with the Cary Varian UV Scan Application (v3.001339). The absorption peak at 483 nm was used to calculate the doxorubicin loading in the DOX-polymer samples. Absorbance measurements were performed in the 200–800 nm range, and baseline correction for PBS was applied.

The samples for transmission electron microscopy (TEM) imaging were prepared by the following method: One drop of solution (0.05 mg·mL$^{-1}$) was cast onto a 300-mesh Formvar® carbon-coated copper TEM grid (Electron Microscopy Sciences, Hatfield, PA, USA, FCF300-Cu) placed onto filter paper and excess solution was wicked off with filter paper. After 1 min, one drop of 2% (*w/v*) phosphotungstic acid was added to the grid, and the excess staining solution was wicked off with filter paper. Finally, the grid was transferred onto a new piece of filter paper in a Petri dish and left to dry overnight at room temperature. The micelles were imaged with a Philips (Eindhoven, The Netherlands) CM10 electron microscope operated at 60 kV acceleration voltage. The images were recorded with an Advance Microscopy Techniques 11-megapixel digital camera and the Image Capture Software Engine version 5.42.558. The feature sizes and size distributions were measured with the open-source processing program ImageJ (version 1.46r) [24]. A minimum of 15 measurements were taken for each sample to provide adequate average size information. In some cases, contrast adjustment was necessary to improve visualization and help with the feature measurements.

Atomic force microscopy (AFM) images were recorded in the tapping mode on a Nanoscope III instrument (model MMAFM-2, scan stage J, Digital Instruments, Santa Barbara, Ca, USA) housed in a NanoCube acoustic isolation cabinet (Novascan, Boone, IA, USA) and mounted on a Halcyonics Micro 40 vibration isolation table (Novascan). The polymer solutions were prepared at concentrations ranging from 0.01 to 0.05 mg·mL$^{-1}$. A 20 μL aliquot of the solution was deposited on the mica substrate and spun at about 3000 revolutions per minute (rpm) for 60 s under ambient conditions. The measurements were performed using Si probes (VistaProbes, T300, NanoScience Instruments, Phoenix, AZ, USA) having a spring constant of 40 N/m, a resonance frequency of 300 kHz, and the following characteristics: length 125 μm, width 40 μm, tip height 14 μm, and tip radius < 10 nm. The images were analyzed using the Nanoscope v 1.40 software. The scan rate was typically between 0.7 and 1.2 Hz at a scan angle of 0°, acquiring 512 samples/line.

### 2.3. Synthesis

The synthesis of the monomer γ-benzyl L-glutamic acid *N*-carboxyanhydride (Glu-NCA), the PBG linear side chains, the linear PBG substrates with different lengths, the partially and fully deprotected linear PBG substrates, the partially deprotected arborescent PBG cores of generation 0 (G0) and generation 1 (G1), amine-terminated linear poly(ethylene oxide) (PEO) with $M_n$ = 10,100 g/mol, and their characterization were accomplished according to procedures described previously [25]. Details on the synthesis and the characterization of PEO are provided as Supplementary Materials (SM).

The different linear PBG substrates are identified in the form PBG$_x$, where X denotes the experimental number of PBG repeating units in the chains. The arborescent G0PBG molecules obtained by coupling the amine-terminated PBG side chains with the different carboxyl-functionalized PBG substrates are likewise identified as G0PBG$_x$, where X denotes the number of repeating units in the substrate used for the synthesis of G0PBG. For example, G0PBG$_{29}$ represents an arborescent G0PBG derived from the PBG substrate with 29 repeating units. Similar sample notation is also used for the G1 and G2 samples.

Linear PBG substrates with different molecular weights and deprotection levels were used in the synthesis of arborescent G0PBG, while the PBG side chains (core building blocks) used in the subsequent reactions had $M_n \approx 5000$ g/mol. The three linear PBG substrates used had an increasing number of PBG repeating units ($X_n$ = 15, 29, and 65,

identified as PBG$_{15}$, PBG$_{29}$, and PBG$_{65}$, respectively). Their deprotection level also varied from 31% (PBG$_{29}$ and PBG$_{65}$) to 100% (PBG$_{15}$) of free glutamic acid moieties [25].

The synthesis of linear Glu(OtBu)$_2$-Poly($\gamma$-benzyl L-glutamate), (*t*BuO)$_2$-PBG), serving as side chains for the last grafting cycle of the arborescent PBG core, yielding the chain-end functionalized G1 and G2 arborescent PBG substrates were obtained as described previously [23]. Detailed information on the synthesis and characterization of these materials is provided as Supplementary Materials (SM).

### 2.4. Preparation of DOX-Loaded Unimolecular Micelles

DOX·HCl (2 mg) was first dissolved in 0.5 mL of DMSO and neutralized with two equivalents of TEA (4 μL) to obtain the drug in its free base (hydrophobic) form. The arborescent copolymer (10 mg) was dissolved in 1 mL of DMSO and stirred for 2 h. Then, the DOX solution was added, and the mixture was stirred overnight in the dark. The organic solvent and the free drug were removed by dialysis (MWCO 3500) against deionized water (1 L) for 24 h with three changes of the dialysis medium, and the resulting solution was either lyophilized in the dark or used directly for the measurements. For the determination of the drug loading content (DLC) and the drug loading efficiency (DLE), the lyophilized DOX-loaded micelles (3 mg) were dissolved in deionized water, and the absorbance was measured on a UV-Vis spectrometer at 483 nm. The DLC and DLE were calculated using the following equations:

$$\text{DLC} = \frac{\text{mass of drug in micelles}}{\text{mass of micelles and drug}} \times 100\% \qquad (1)$$

$$\text{DLE} = \frac{\text{mass of drug in micelles}}{\text{total mass of drug in feed}} \times 100\% \qquad (2)$$

### 2.5. In Vitro Release of DOX

To determine the DOX release profiles, a 3 mg sample of freeze-dried DOX-loaded micelles was dispersed in 1 mL of phosphate-buffered saline (release medium, 10 mM phosphate, 137 mM NaCl, and 2.7 mM KCl, pH 7.4 or 5.5; the pH was adjusted to 5.5 with 6 M HCl) and transferred to a dialysis bag (MWCO 3500). The release experiment was initiated by placing the sealed dialysis bag into 3 mL of release medium at the same pH and 37 °C, with constant shaking at 100 rpm. At selected time intervals, the release medium was completely withdrawn and replaced with 3 mL of fresh medium. The amount of DOX released was determined from the absorbance measured on a UV-Vis spectrometer at 483 nm and a calibration curve. The drug release studies were performed in triplicate for each sample.

### 3. Results and Discussion

The structure of arborescent copolymers can be tailored to construct unimolecular micelles for various applications, particularly for drug delivery. For example, the branching density (porosity) of the molecules can be varied by adjusting the length of the chains and the functionalization (deprotection) level of the arborescent PBG substrates, while their overall size and branching functionality can be controlled by using substrates of different generations. Linear PBG substrates with different molecular weights and deprotection levels were used in the current investigation for the synthesis of arborescent G0PBG, while the PBG side chains used in the subsequent reactions all had M$_n$ ≈ 5000 g/mol. The three linear PBG substrates used had an increasing number of PBG repeating units (X$_n$ = 15, 29, and 65, identified as PBG$_{15}$, PBG$_{29}$, and PBG$_{65}$, respectively). The number of PBG units was controlled by varying the ratio of monomer ($\gamma$-benzyl L-glutamic acid N-carboxyanhydride, Glu-NCA) to the initiator (*n*-hexylamine) used in the polymerization reaction. Their deprotection level was also varied from 31 mol% (PBG$_{29}$ and PBG$_{65}$) to 100 mol% (PBG$_{15}$) of free glutamic acid moieties. As a result, three well-defined (polydis-

persity index PDI = $M_w/M_n$ < 1.09) comb-branched or arborescent G0PBG were obtained (Scheme 2A), but with a structure varying from compact and dense for G0PBG$_{15}$ to a loose and more porous structure for G0PBG$_{65}$, and an intermediate (semi-compact) structure for G0PBG$_{29}$. The different substrate lengths and deprotection levels of linear PBG were used initially in the synthesis of G0PBG in the hope that the topology (shape) of the molecules could be varied from spherical to rod-like, but TEM and AFM imaging indicated that all the molecules were spherical. This approach nevertheless allowed variations in the branching density (porosity) for the cores, which seems to correlate well with the release properties of the molecules (as discussed below). A detailed discussion of the synthesis of these linear PBG substrates and the arborescent G0PBG serving as hydrophobic core components for the synthesis of arborescent copolymers was provided previously [25].

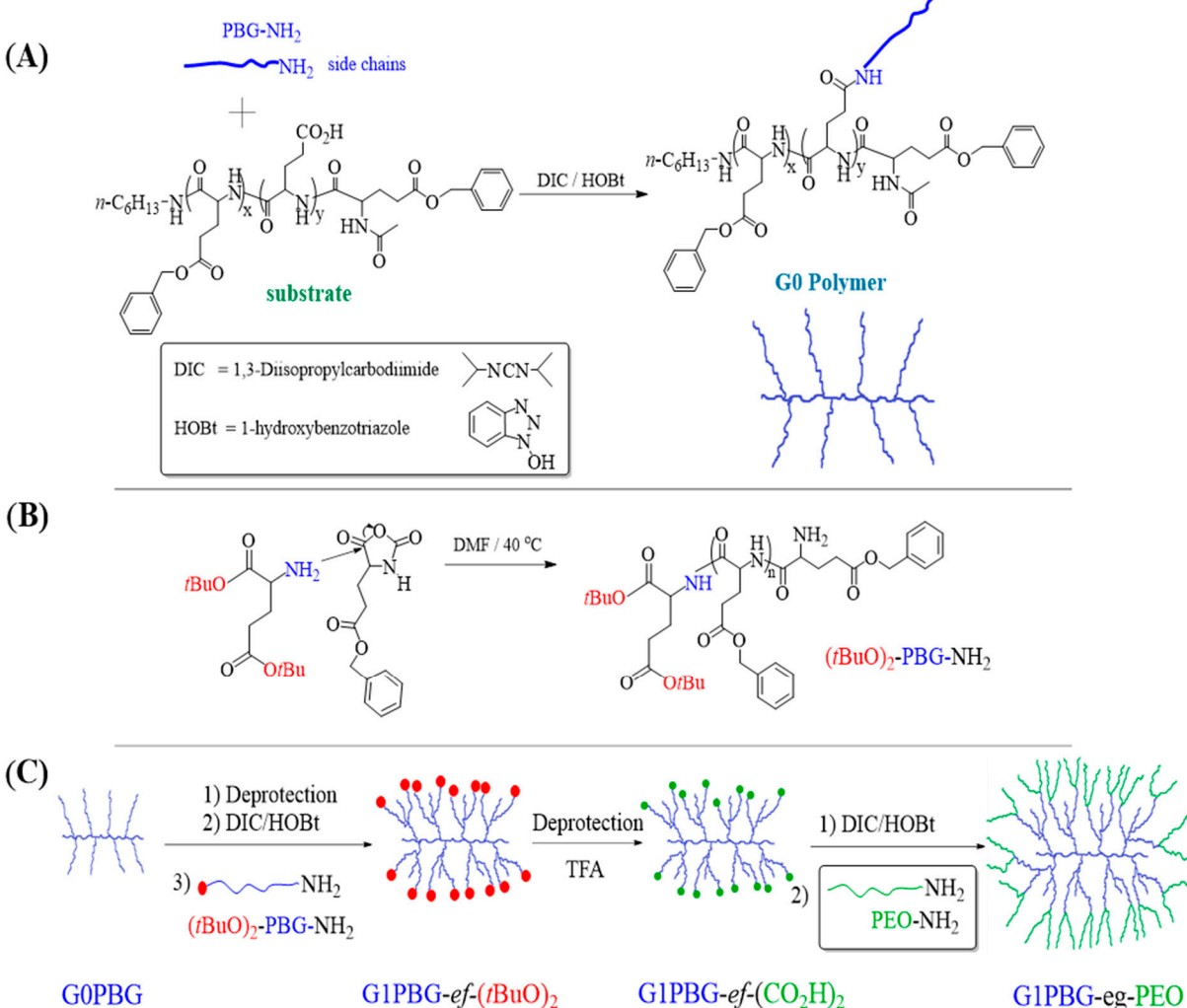

**Scheme 2.** (**A**) Synthesis of a G0 arborescent polymer from PBG building blocks; the subscripts x and y refer to the mole fractions of protected and deprotected benzyl glutamate units on the linear PBG substrates, respectively. The three linear PBG substrates used had number-average degrees of polymerization X$_n$ = 15, 29, and 65, identified as PBG$_{15}$, PBG$_{29}$, and PBG$_{65}$, respectively. Their deprotection level "y" varied from 31% (PBG$_{29}$ and PBG$_{65}$) to 100% (PBG$_{15}$). (**B**) Polymerization of γ-benzyl L-glutamic acid N-carboxyanhydride (Glu-NCA) using Glu(O*t*Bu)$_2$·HCl as initiator. (**C**) Schematic representation of the synthesis of chain end carboxyl-functionalized G1PBG substrate (G1PBG-*ef*-(CO$_2$H)$_2$) and the chain end-grafted arborescent copolymer G1PBG-*eg*-PEO.

The characterization data for the linear PBG and arborescent G0PBG substrates are summarized in Tables S1 and S2 (Supplementary Materials (SM)), respectively. As seen

in Table S2, the number-average branching density ($b_d$), defined as the number of side chains added in the grafting reaction divided by the number of repeating units on the linear chain substrate, increased as the chain length of the substrate decreased. G0PBG$_{15}$ had the highest branching density ($b_d$ = 0.80) among the three PBG substrates and therefore should have a more compact and dense core structure. In contrast, G0PBG$_{65}$ had the lowest branching density ($b_d$ = 0.11) and should have a more porous core structure, while G0PBG$_{29}$-*eg*-PEO/DOX ($b_d$ = 0.19) is intermediate.

### 3.1. Synthesis of Linear (tBuO)$_2$-PBG and Arborescent PBG Substrates with Carboxylic Acid Chain Ends

Linear PBG with two *tert*-butyl ester protecting groups at one chain end, (*t*BuO)$_2$-PBG, was obtained via the ring-opening polymerization of Glu-NCA initiated with the HCl salt of glutamic acid di-*tert*-butyl ester [23]. The polymerization of NCA monomers initiated with primary amine hydrochlorides (rather than free amines) has been shown to be beneficial in avoiding termination reactions, but it is also slower due to decreased reactivity of the initiator and the propagating centers [26–28]. Consequently, the polymerization was carried out in DMF at 40 °C for 7 days (Scheme 2B). Detailed information on the synthesis and the characterization of (*t*BuO)$_2$-PBG is provided as Supplementary Materials (SM). The number-average degree of polymerization ($X_n$) of the (*t*BuO)$_2$-PBG sample was determined by $^1$H NMR analysis, as shown in Figure S1, by comparing the integrated peak intensities for the benzylic methylene protons in the repeating units (2H at 4.9 ppm) and the protons of the *tert*-butyl ester groups in the initiator fragment (18H at 1.3 ppm). The polymerization initiated with Glu(OtBu)$_2$·HCl yielded PBG with $X_n$ = 24, in good agreement with the target $X_n$ = 25.

Chain end-functionalized arborescent substrates were obtained by grafting (*t*BuO)$_2$-PBG onto randomly deprotected arborescent PBG substrates of generations G0 and G1. The deprotection level of the PBG substrates was ca. 30%, and a 1:1.1 molar ratio of side chains NH$_2$ chain ends to CO$_2$H groups on the substrate was used to maximize the grafting yield and the coupling efficiency (Table S3). The unreacted side chains were removed from the crude polymer by preparative size exclusion chromatography (SEC). The absolute molecular weight of the arborescent PBG samples was determined by SEC-LS analysis in DMSO. The molecular weight and the branching functionality ($f_n$) of the polymers increased with the generation number as expected while maintaining a low polydispersity index (PDI $\leq$ 1.09). The lower grafting yield ($G_y$) and coupling efficiency ($C_e$) observed for sample G1PBG$_{15}$ are attributed to the dense structure of the G0PBG$_{15}$ substrate, making it difficult for the linear side chains to diffuse to the coupling sites.

### 3.2. Synthesis of Chain End-Grafted Arborescent Copolymers

Previously, we randomly grafted linear poly(ethylene oxide) (PEO) with $M_n$ = 10,100 g/mol onto arborescent PBG substrates to generate arborescent copolymer micelles as nanocarriers for doxorubicin (DOX) [25]. Unfortunately, the randomly grafted arborescent PBG-*g*-PEO copolymers obtained by that approach yielded a small number of aggregated species in phosphate-buffered saline (PBS). We hypothesized that by grafting PEO selectively at the end of the chains on arborescent PBG substrates, as shown in Figure 1, the solubility of the copolymers in PBS could be enhanced to yield stable unimolecular micelles without aggregation. Since the coupling sites for the chain end-functionalized PBG substrates are located closer to the periphery of the molecules, a better-defined crew-cut core–shell morphology is expected for the amphiphilic copolymers. Consequently, the hydrophilic PEO shell should be able to shield the hydrophobic PBG core from intermolecular hydrophobic interactions and the aqueous environment more efficiently, leading to stable unimolecular micelles free of aggregation.

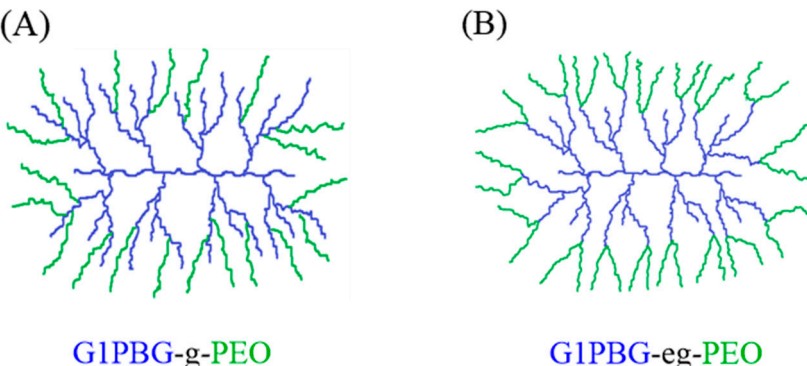

**Figure 1.** Comparison of the structure of (**A**) randomly and (**B**) chain end-grafted arborescent copolymers derived from G1 PBG substrates.

The arborescent PBG substrates with *tert*-butyl ester-protected carboxylic acid groups at the chain ends were dissolved in trifluoroacetic acid (TFA) to selectively cleave the *tert*-butyl ester-protecting groups and generate two carboxylic acid groups at each chain end (PBG-*ef*-(CO$_2$H)$_2$). These chain end-functionalized arborescent PBG samples served as substrates in a subsequent grafting reaction with amine-terminated PEO chains to obtain the chain end-grafted amphiphilic arborescent copolymers PBG-*end-grafted*-PEO (PBG-*eg*-PEO) as shown in Scheme 2.

The mole fraction of *tert*-butyl ester groups in the arborescent substrates was determined by $^1$H NMR analysis, as shown in Figure 2A, by comparing the integrated peak intensities for the methine protons in the repeating units (1H at 3.9 ppm) and the protons of the *tert*-butyl ester groups in the initiator fragment (18H at 1.3 ppm). A calculation of the mole fraction of *tert*-butyl ester groups before deprotection of the G1 PBG sample, G1PBG15-*ef*-(*t*BuO)$_2$, is provided in Equation (3) as an example.

$$f_{tBu} = \frac{(\text{integral } tert-\text{butyl H})/18}{(\text{integral methine H})/1} = \frac{0.82/18}{1/1} = 0.045 \times 2 = 0.09 \tag{3}$$

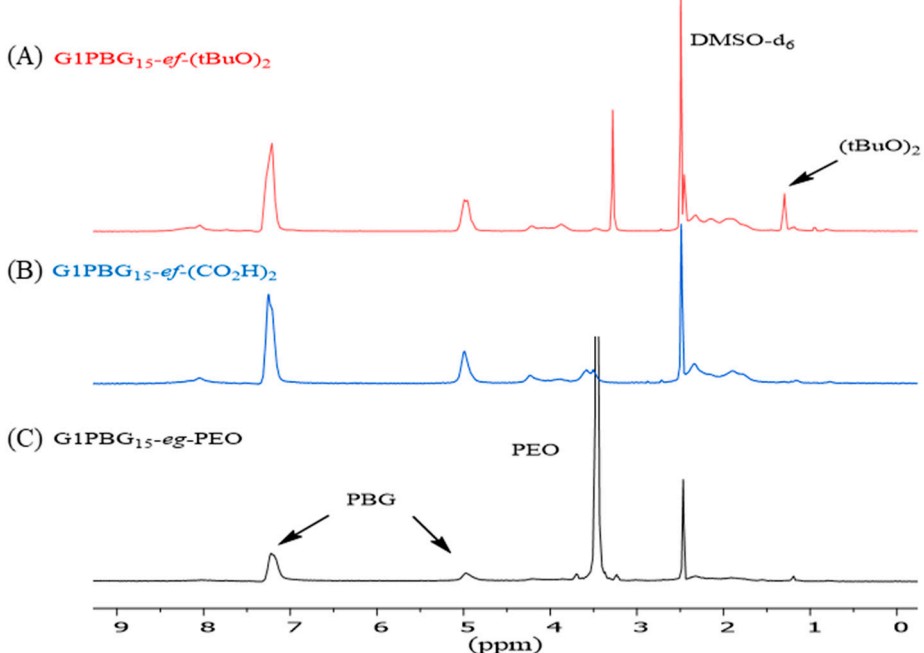

**Figure 2.** $^1$H NMR spectra for G1PBG$_{15}$-*ef*-(*t*BuO)$_2$ (**A**) before and (**B**) after deprotection of the *tert*-butyl ester groups and for G1PBG$_{15}$-*eg*-PEO (**C**) in DMSO-$d_6$.

$^1$H NMR analysis was also used to confirm that complete deprotection of the *tert*-butyl ester groups was achieved before grafting PEO onto the PBG substrate. The complete disappearance of the *tert*-butyl protons at 1.3 ppm confirmed their complete removal (Figure 2B).

The results obtained when coupling the six different G1 and G2 chain end-functionalized PBG-*ef*-(CO$_2$H)$_2$ substrates with PEO side chains to yield the corresponding arborescent PBG copolymers end-grafted with PEO (PBG-*eg*-PEO) are summarized in Table 1. The absolute molecular weight of the substrates and the arborescent copolymers was determined by SEC-LS analysis in DMSO. In all cases, the molecular weight of the copolymers increased with respect to the PBG substrates while maintaining a low polydispersity index (PDI $\leq$ 1.09).

**Table 1.** Characteristics of chain end-grafted PBG-*eg*-PEO arborescent copolymers.

| Copolymer | PBG Substrate | | | Graft Copolymer | | | |
| | $M_n$ (g/mol) [a] | mol % −CO$_2$H [b] | $G_y$ [c] | $M_n$ (g/mol) [a] | $M_w/M_n$ [a] | $f_n$ [d] | wt% PEO [e] |
|---|---|---|---|---|---|---|---|
| G1PBG$_{15}$-*eg*-PEO | 220,000 | 9 | 70 | 890,000 | 1.06 | 67 | 75 |
| G1PBG$_{29}$-*eg*-PEO | 210,000 | 8 | 68 | 760,000 | 1.09 | 55 | 72 |
| G1PBG$_{65}$-*eg*-PEO | 270,000 | 9 | 63 | $1.0 \times 10^6$ | 1.05 | 73 | 73 |
| G2PBG$_{15}$-*eg*-PEO | $1.2 \times 10^6$ | 9 | 24 | $2.4 \times 10^6$ | 1.05 | 120 | 50 |
| G2PBG$_{29}$-*eg*-PEO | 970,000 | 9 | 31 | $2.3 \times 10^6$ | 1.09 | 133 | 57 |
| G2PBG$_{65}$-*eg*-PEO | $1.3 \times 10^6$ | 7 | 20 | $2.2 \times 10^6$ | 1.07 | 90 | 40 |

[a] values from SEC-LS analysis in DMSO; [b] mole fraction of carboxyl groups in the substrate; [c] grafting yield: fraction of side chains becoming attached to the substrate in the grafting reaction; [d] branching functionality: number of branches added in the last grafting cycle; [e] PEO weight fraction, from the difference in $M_n$ of the copolymer and the substrate.

The grafting yield ($G_y$), defined as the fraction of linear chain segments becoming attached to the substrate, was calculated by the same method described for the randomly grafted systems [25]. While a lower functionality level of CO$_2$H coupling sites was achieved for the chain end-functionalized PBG substrates (7–9 mol %) in comparison to randomly functionalized PBG substrates (30–33 mol %) [25], higher grafting yields were obtained under the same reaction conditions. For example, the PEO grafting yield for chain end grafting was 70, 68, and 63% for G1PBG$_{15}$-*eg*-PEO, G1PBG$_{29}$-*eg*-PEO, and G1PBG$_{65}$-*eg*-PEO, respectively, significantly higher than for the randomly grafted systems (27, 24, and 29% for G1PBG$_{15}$-*g*-PEO, G1PBG$_{29}$-*g*-PEO, and G1PBG$_{65}$-*g*-PEO, respectively [25]). This can be explained by the better accessibility of the coupling sites in the chain end-functionalized PBG substrates in comparison with their randomly functionalized analogs. Despite the better accessibility of the coupling sites in the chain end-functionalized PBG substrates, arborescent copolymers with lower molar masses were obtained for the G1 and G2 polymers with different PBG core structures relative to the randomly functionalized analogs. This means that a lower number of hydrophilic PEO segments were added in the last grafting cycle, and thus a lower branching functionality $f_n$ was achieved. This is obviously a consequence of the lower functionality level of the chain end-functionalized PBG substrates.

SEC traces obtained for the purified copolymer samples are compared in Figure 3, and the analysis results are summarized in Table 1. A decrease in elution volume is observed as the generation number increases. The elution volume also increases in the G1 polymer series as G1PBG$_{65}$-*eg*-PEO < G1PBG$_{15}$-*eg*-PEO < G1PBG$_{29}$-*eg*-PEO, while for the G2 series, it increases as G2PBG$_{15}$-*eg*-PEO < G2PBG$_{29}$-*eg*-PEO < G2PBG$_{65}$-*eg*-PEO. The variations in absolute $M_n$ within each series (Table 1) are small, however, with average values of ca. $(9 \pm 1) \times 10^5$ g/mol and $(2.3 \pm 0.1) \times 10^6$ g/mol for the G1 and G2 copolymers, respectively.

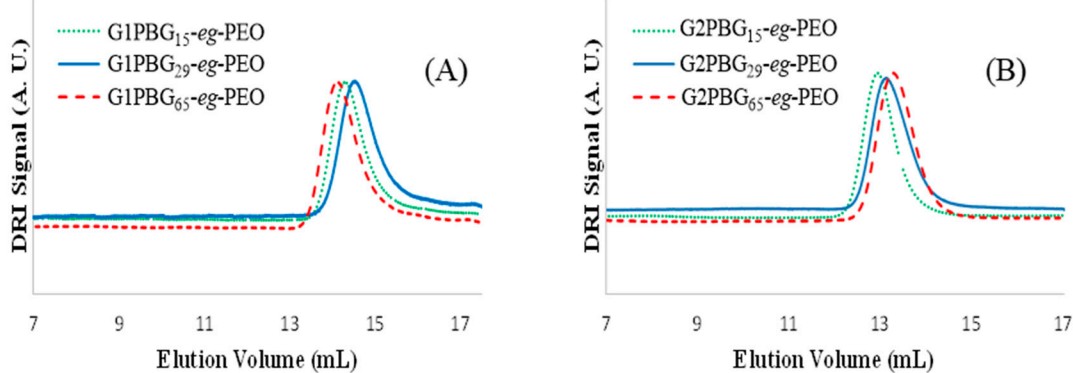

**Figure 3.** SEC traces for (**A**) G1 and (**B**) G2 arborescent copolymers.

### 3.3. Properties of Chain End-Grafted Arborescent Copolymer Micelles

The chain end-grafted arborescent copolymers were investigated using dynamic light scattering (DLS), transmission electron microscopy (TEM), and atomic force microscopy (AFM) measurements to examine the influence of the dendritic core structure on the micelle properties. The hydrodynamic diameter determined by DLS followed the expected increasing trend from G1 to G2 molecules (Table 2). Whitton and Gauthier previously reported the use of linear PEO with a molecular weight of 5000 g/mol, grafted either randomly or at the end of the chains on arborescent PBG substrates, to generate arborescent copolymer micelles. Unfortunately, the randomly grafted copolymers of generations G1 obtained by that approach yielded large aggregated species in PBS, while generations G2 and G3 were both insoluble. The corresponding end-grafted samples PBG-*eg*-PEO displayed better dispersibility in PBS than the randomly grafted systems, albeit a small population of aggregates still existed in these samples [23]. In the current investigation, we expanded this approach by designing amphiphilic arborescent copolymers having PEO chain segments with $M_n$ = 10,100 g/mol end-grafted onto arborescent PBG substrates of generations G1 and G2. The addition of these hydrophilic PEO segments strictly at the chain ends of the arborescent hydrophobic PBG cores clearly facilitated the formation of unimolecular micelles without aggregates in aqueous solutions, in contrast to the randomly grafted copolymers investigated previously [25].

**Table 2.** Characteristics of chain end-grafted arborescent PBG-*eg*-PEO copolymers.

| Copolymer | DMF | | | | PBS | | | |
|---|---|---|---|---|---|---|---|---|
| | $D_{h, number}$ | $D_{h, volume}$ | $D_{h, intensity}$ | PDI | $D_{h, number}$ | $D_{h, volume}$ | $D_{h, intensity}$ | PDI |
| G1PBG$_{15}$-*eg*-PEO | 11 ± 1 | 15 ± 1 | 17 ± 1 | 0.08 | 13 ± 2 | 17 ± 2 | 26 ± 2 | 0.19 |
| G1PBG$_{29}$-*eg*-PEO | 12 ± 1 | 13 ± 1 | 15 ± 1 | 0.10 | 14 ± 2 | 17 ± 2 | 22 ± 2 | 0.31 |
| G1PBG$_{65}$-*eg*-PEO | 14 ± 2 | 16 ± 2 | 18 ± 2 | 0.12 | 14± 2 | 16 ± 2 | 19 ± 3 | 0.27 |
| G2PBG$_{15}$-*eg*-PEO | 25 ± 2 | 27 ± 2 | 32 ± 2 | 0.04 | 29 ± 1 | 33 ± 1 | 37 ± 1 | 0.19 |
| G2PBG$_{29}$-*eg*-PEO | 25 ± 1 | 29 ± 1 | 35 ± 1 | 0.05 | 26 ± 1 | 31 ± 3 | 39 ± 1 | 0.27 |
| G2PBG$_{65}$-*eg*-PEO | 21 ± 1 | 23 ± 1 | 29 ± 1 | 0.09 | 22 ± 1 | 27 ± 1 | 31 ± 1 | 0.23 |

All the experiments were carried out in triplicate, and the diameters are presented as the mean ± standard deviation (nm) for three measurements.

The good agreement in the hydrodynamic diameter of the copolymers determined by DLS analysis in both DMF (a good solvent for the core and shell components) and in PBS solution (an aqueous environment, selective for the PEO shell) indeed suggests that unimolecular micelles were obtained with number-average mean diameters of 11–14 and 21–29 nm for the G1 and G2 molecules, respectively (Table 2). Similar trends were also observed for the volume- and intensity-average diameters. The average hydrodynamic diameter of the end-grafted samples is clearly lower than of their randomly grafted analogs [25].

Size distribution analysis of the DLS results for the G1PBG-*eg*-PEO copolymers confirmed that the chain end-grafted copolymers formed exclusively stable unimolecular species without aggregation in DMF. Analysis of the same copolymers in phosphate-buffered saline (PBS) solution yielded similar results, with number-average diameters comparable to those obtained in DMF in most cases. Interestingly, analysis of the G1PBG-*eg*-PEO copolymers yielded a very small amount (1–3%) of aggregates with a diameter around 5000 nm in the intensity-weighted distributions, while no aggregation was detected in the number-weighted nor in the volume-weighted distributions (Figure S3). This is attributed to insignificant aggregation in these samples. The DLS measurements for all the G2PBG-*eg*-PEO samples yielded no detectable amounts of aggregates in DMF nor in PBS, as shown in Figure 4. The absence of aggregates in these systems clearly shows that the better-defined crew-cut core–shell morphology of chain end-grafted arborescent copolymers is a key factor in the formation of unimolecular micelles. In spite of the different hydrophobic G0PBG core structures used in the synthesis of the chain end-grafted G2 copolymers, only minor variations in hydrodynamic diameter were obtained in DLS analysis. This is attributed to the collapse of the hydrophobic PBG cores to minimize their exposure to the aqueous environment (as confirmed by [1]H NMR analysis below), leading to a comparable contribution to the overall size of the micelles.

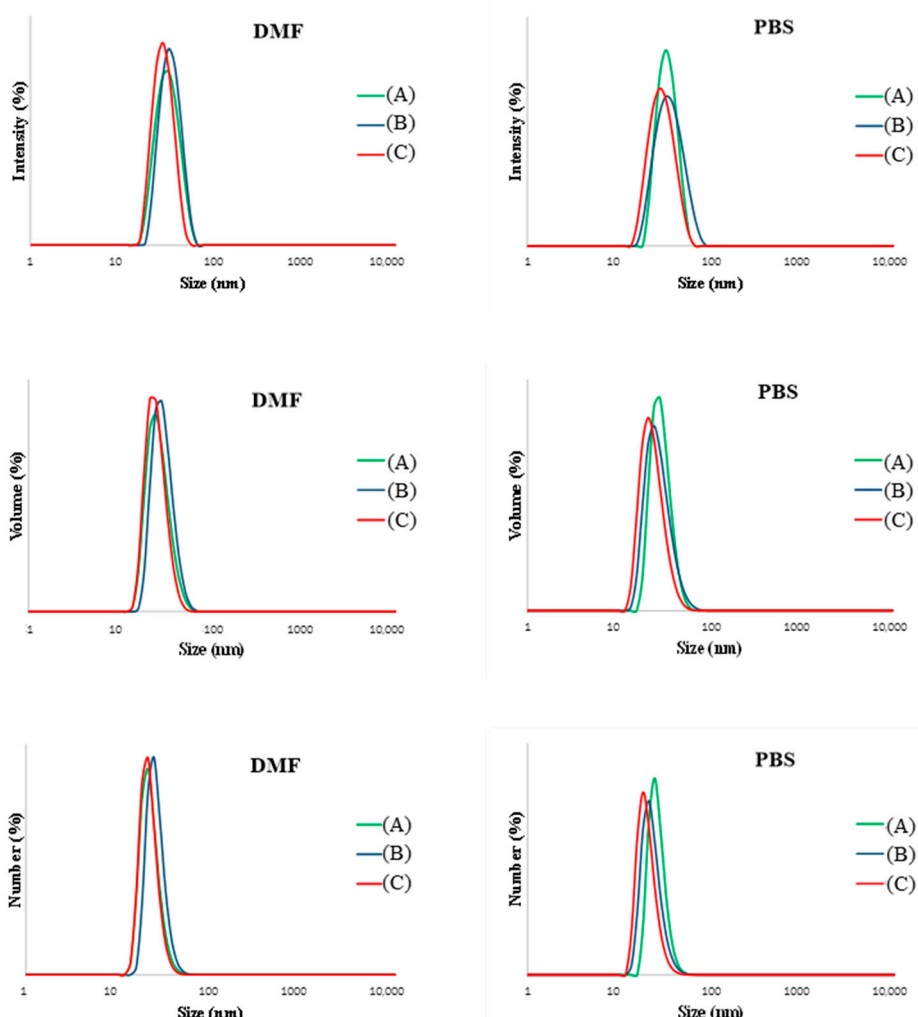

**Figure 4.** Hydrodynamic diameter distributions of the arborescent copolymers determined by DLS in DMF and in PBS solution: (**A**) G2PBG$_{15}$-*eg*-PEO, (**B**) G2PBG$_{29}$-*eg*-PEO, and (**C**) G2PBG$_{65}$-*eg*-PEO.

The formation of micelles with a collapsed core structure was further confirmed via [1]H NMR spectroscopy analysis (Figure S4). For example, the analysis of G2PBG$_{15}$-*eg*-PEO

in D$_2$O displayed no resonances for the phenyl (5H at 7.3) and benzylic methylene protons (2H at 4.9 ppm), while these signals were present in DMSO-*d$_6$*, a good solvent for both the PBG core and the PEO sides chains, indicating that the G2PBG core was collapsed in D$_2$O.

Further analysis of the chain end-grafted copolymers via TEM (after staining with phosphotungstic acid) and AFM confirmed that all the samples had a uniform size distribution and a spherical shape (Figure S5 for G1 copolymers; Figure 5 for G2 copolymers). The average diameter measured by TEM for the G1 copolymers was 10 ± 2, 10 ± 3, and 13 ± 3 nm for G1PBG$_{15}$-*eg*-PEO, G1PBG$_{29}$-*eg*-PEO, and G1PBG$_{65}$-*eg*-PEO, respectively. For the G2 copolymers, the average diameter was 23 ± 3, 21 ± 5, and 18 ± 4 nm for G2PBG$_{15}$-*eg*-PEO, G2PBG$_{29}$-*eg*-PEO, and G2PBG$_{65}$-*eg*-PEO, respectively. The TEM measurements therefore confirm the trends observed by DLS analysis of increasing micelle size with the generation number, but the diameter is smaller because TEM analysis was performed in the dry state. Although it was expected that the different PBG linear substrates would yield micelles with different shapes, only spherical micelles were observed in both the AFM and TEM images (Figures S4 and 5) for all the samples. While the exact reason for this is unknown, several factors may have contributed to the similar shape of the micelles, including the branched structure of the copolymers, solvent effects, and intramolecular hydrophobic interactions [29–31].

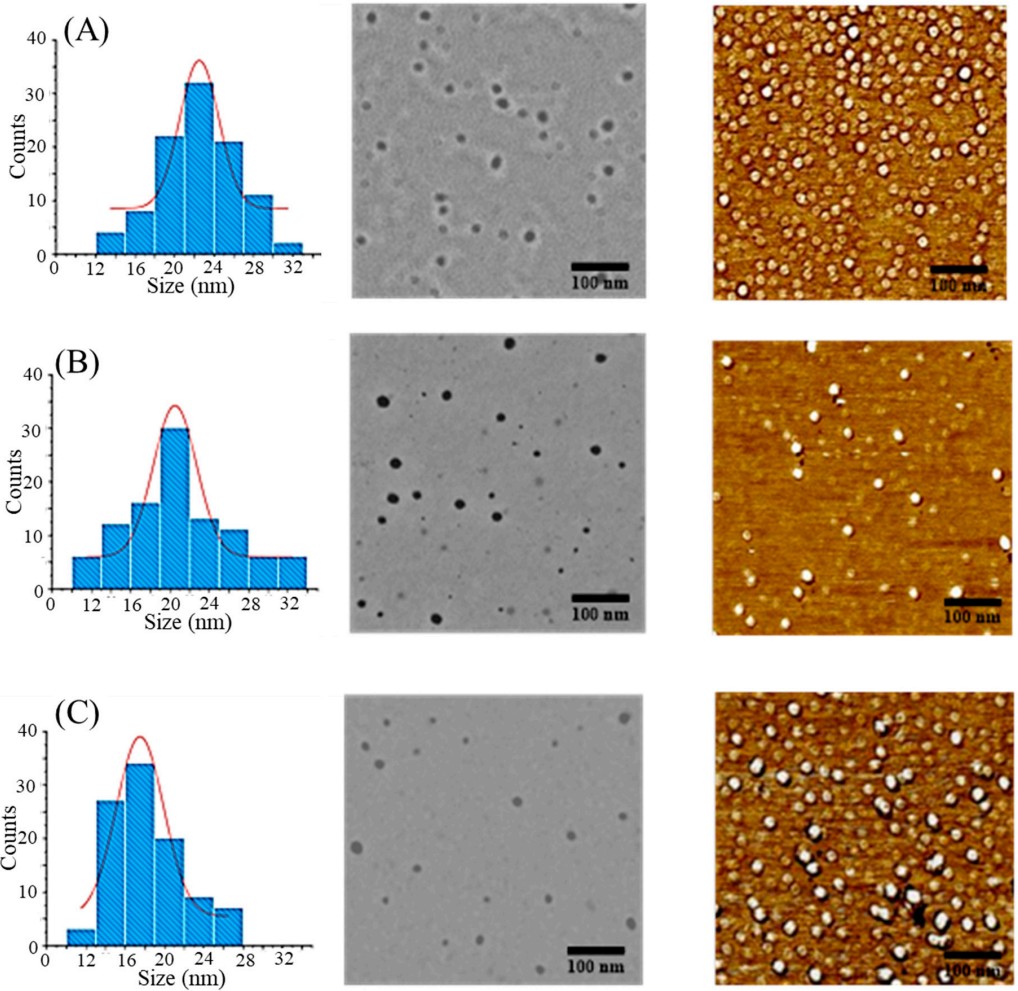

**Figure 5.** TEM size distribution histograms with Gaussian-fitting curves (solid lines) (**left**), TEM (**middle**), and AFM phase (**right**) imaging for chain end-grafted arborescent copolymers: (**A**) G2PBG$_{15}$-*eg*-PEO, (**B**) G2PBG$_{29}$-*eg*-PEO, and (**C**) G2PBG$_{65}$-*eg*-PEO.

### 3.4. Drug Loading and Micelle Characterization

The arborescent copolymers synthesized possess an amphiphilic core–shell morphology with a hydrophobic arborescent poly(γ-benzyl L-glutamate) (PBG) core, creating a favorable microenvironment for the encapsulation of hydrophobic drugs via physical entrapment, and a hydrophilic shell of poly(ethylene oxide) (PEO) making these copolymers water-soluble (Scheme 3). The salt form of DOX (DOX·HCl) is water-soluble, and it can interact with anionic polymers via electrostatic interactions. However, even in that case, the interactions can be disrupted by small changes in pH [32]. To favor encapsulation via hydrophobic interactions, the hydrophilic doxorubicin hydrochloride (DOX·HCl) was neutralized with two equivalents of triethylamine (TEA) to obtain the more hydrophobic DOX free base before encapsulation, allowing the drug to partition into the core of the micelles as shown in Scheme 3 [33,34].

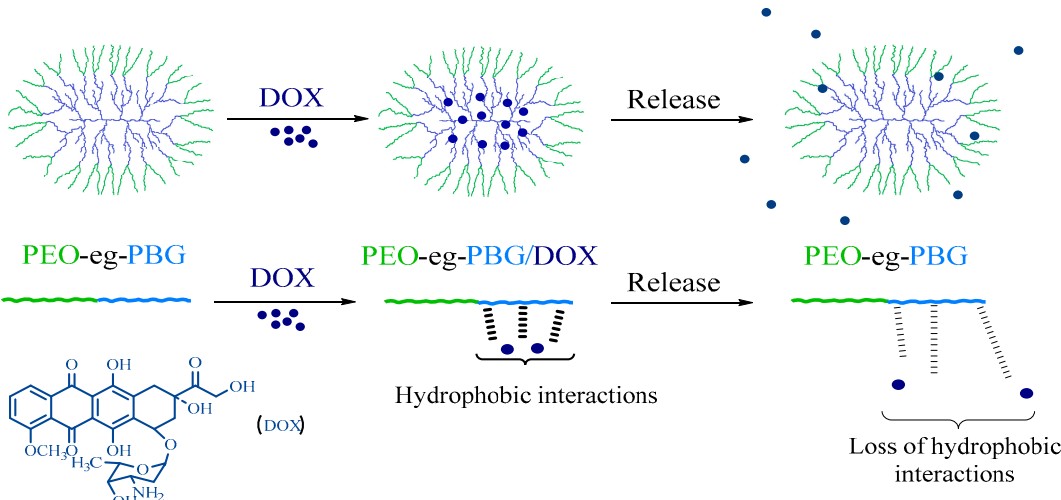

**Scheme 3.** Encapsulation and release of DOX in the hydrophobic PBG core of G2PBG-*eg*-PEO via hydrophobic interactions.

In contrast to amphiphilic block copolymer micelles, unimolecular micelles are stable irrespective of their concentration or the solvency conditions used. Unimolecular micelles do not dissociate upon dilution, for example, in the bloodstream, and thus should lead to more controlled drug release and increased treatment efficacy. To evaluate the encapsulation and release properties of the synthesized arborescent copolymers, doxorubicin (DOX) was used as a model hydrophobic drug. The influence of differences in core structure on the ability to encapsulate and control the release of the hydrophobic drug was investigated.

The encapsulation study confirmed that following purification by dialysis, DOX could be solubilized in aqueous solutions of amphiphilic arborescent copolymers of generations G1 and G2. For example, the encapsulation of DOX within the hydrophobic core of G2PBG-*eg*-PEO yielded enhanced absorption around 490 nm for G2PBG-*eg*-PEO/DOX as compared with the blank micelles (Figure 6). The UV–vis spectra exhibited a bathochromic shift for the encapsulated drug ($\lambda_{max}$ = 490 nm) as compared to free DOX ($\lambda_{max}$ = 483 nm), attributed to hydrophobic interactions with the PBG core, which confirms successful DOX encapsulation within the micelles.

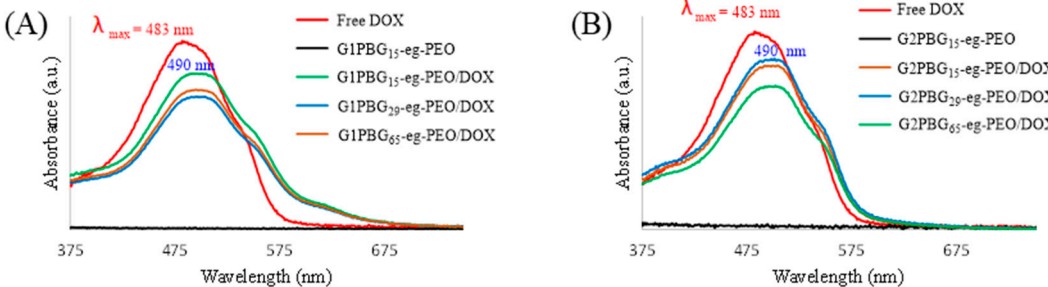

**Figure 6.** UV Absorption in PBS of free DOX and (**A**) G1, (**B**) G2 arborescent copolymers.

[1]H NMR spectroscopy was also used to confirm the encapsulation of DOX, as shown in Figure S6, by complete disappearance of the PBG and DOX signals for G2PBG$_{15}$-*eg*-PEO/DOX in D$_2$O (Figure S6c), while resonances were observed for free DOX in D$_2$O at the same overall concentration of 2 mg/mL (Figure S6a), indicating restricted mobility of the hydrophobic PBG core and the DOX molecules entrapped within the core of the nanocarriers.

The physical entrapment of hydrophobic drugs in micelles is mainly driven by hydrophobic interactions between the drug and the core of the micelles. By increasing the PBG content from a G1PBG to a G2PBG substrate in the unimolecular micelles, it should be possible to load more DOX in the micelles. Indeed, the DOX loading capacity of the arborescent copolymers was higher for all the G2PBG-*eg*-PEO/DOX samples as compared with the G1PBG-*eg*-PEO/DOX systems (Table 3). For example, sample G2PBG$_{15}$-*eg*-PEO/DOX had higher values of DLC (11.2%) and DLE (67%) as compared to sample G1PBG$_{15}$-*eg*-PEO/DOX, with DLC and DLE values of 7.9% and 47%, respectively. This is attributed to the higher PBG content of G2PBG-*eg*-PEO/DOX relative to G1PBG-*eg*-PEO/DOX, allowing for more DOX molecules to be encapsulated within the PBG core via hydrophobic interactions. Moreover, the DOX loading capacity of the arborescent copolymers varied slightly within the series G2PBG$_{15}$-*g*-PEO/DOX, G2PBG$_{65}$-*g*-PEO/DOX, G2PBG$_{29}$-*g*-PEO/DOX from 11.2 wt% to 10.4 wt% and 9.7 wt%, respectively. While these variations are minor, they could be related to variations in core size, as seen in Table 3. Increasing the core size led to increased hydrophobicity for the G2PBG substrate in the unimolecular micelles, which enhanced their loading capacity for DOX.

**Table 3.** Characterization of DOX-loaded unimolecular micelles.

| Copolymer | PBG | | PBG-*eg*-PEO | | PBG-*eg*-PEO/DOX | | | |
|---|---|---|---|---|---|---|---|---|
| | $D_h$ (nm) | PDI | $D_h$ (nm) | PDI | $D_h$ (nm) | PDI | DLC (wt%) | DLE (%) |
| G1PBG$_{15}$-*eg*-PEO/DOX | - | - | 13 | 0.19 | 14 | 0.41 | 7.9 ± 0.3 | 47 ± 4 |
| G1PBG$_{29}$-*eg*-PEO/DOX | - | - | 14 | 0.31 | 14 | 0.32 | 7.5 ± 0.2 | 45 ± 4 |
| G1PBG$_{65}$-*eg*-PEO/DOX | - | - | 14 | 0.27 | 15 | 0.35 | 8.1 ± 0.2 | 49 ± 5 |
| G2PBG$_{15}$-*eg*-PEO/DOX | 14 | 0.18 | 29 | 0.19 | 31 | 0.28 | 11.2 ± 0.3 | 67 ± 4 |
| G2PBG$_{29}$-*eg*-PEO/DOX | 10 | 0.05 | 26 | 0.27 | 27 | 0.27 | 9.7 ± 0.2 | 58 ± 3 |
| G2PBG$_{65}$-*eg*-PEO/DOX | 12 | 0.04 | 22 | 0.23 | 24 | 0.29 | 10.4 ± 0.3 | 62 ± 4 |

Diameters represented as the mean ± standard deviation (*n* = 3).

Examining the drug loading efficiency (DLE) to further investigate whether differences in core–shell morphology between the chain end-grafted and randomly grafted arborescent copolymers affect their ability to encapsulate the hydrophobic DOX, it was found that the DLE of all the arborescent copolymers with similar cores did not differ significantly [25]. For example, the end-grafted arborescent copolymers G2PBG$_{15}$-*eg*-PEO encapsulated DOX with a DLE of 67%, while the randomly grafted arborescent copolymers G2PBG$_{15}$-*g*-PEO had a DLE of 65%. The end-grafted copolymer had an insignificantly larger DLE, while

its molar mass and hydrodynamic diameter were significantly lower ($2.4 \times 10^6$ g·mol$^{-1}$, 29 nm, respectively) than for its randomly grafted counterpart ($3.4 \times 10^6$ g·mol$^{-1}$, 35 nm, respectively), which is attributed to the similarity of the core size and hydrophobicity in both copolymers. Similar trends were also observed for the other copolymers. These results further confirm that the branching density ($b_d$) of the hydrophobic core, and thus the core size and the hydrophobicity of the unimolecular micelles, is the main factor affecting their drug loading capacity rather than the overall size of the micelles.

The size of the DOX-loaded micelles in PBS was also investigated by DLS and found to be only slightly larger than the corresponding blank micelles (Table 3). For instance, the number-average diameter of G2PBG$_{15}$-*eg*-PEO increased from 29 to 31 nm upon loading with DOX. This size increase is, of course, small and close to the error limits of the DLS technique (ca. $\pm 1$ nm) but is nevertheless consistent with the presence of entrapped DOX molecules. The following release experiments focused on the G2PBG-*eg*-PEO/DOX systems, which had the highest DLC and DLE.

*3.5. In Vitro Drug Release Kinetics*

The in vitro drug release characteristics of the DOX-loaded micelles were investigated by dialysis at 37 °C in phosphate-buffered saline (PBS) at pH 7.4 (normal physiological conditions) and 5.5 (to mimic tumor cell environments). The release profile for the DOX-loaded micelles at different pH levels is compared with the release profile for free DOX in Figure 7. The pH significantly affected the release kinetics of DOX from all three DOX-loaded micellar systems. At pH 7.4, the DOX release was slower, with a cumulative release after 50 h of only 14.5, 16.5 and 20.4% for G2PBG$_{15}$-*eg*-PEO/DOX (Figure 7A), G2PBG$_{29}$-*eg*-PEO/DOX (Figure 7B), and G2PBG$_{65}$-*eg*-PEO/DOX (Figure 7C), respectively. However, at pH 5.5, the cumulative release under the same conditions increased to 27.8, 28.6 and 39.4% for G2PBG$_{15}$-*eg*-PEO/DOX, G2PBG$_{29}$-*eg*-PEO/DOX, and G2PBG$_{65}$-*eg*-PEO/DOX, respectively. These results confirm that the DOX-loaded micelles show pH-dependent drug release profiles. This is presumably due to the protonation of DOX under acidic conditions, which increases its solubility and accelerates the diffusion rate of DOX. As a result, the hydrophobic interactions between DOX and the core of the micelles are weakened [35,36]. Such pH-dependent release behavior of DOX was pointed out to be highly beneficial in drug delivery for cancer therapy, as it would favor the release of DOX in cancer cells while limiting release in the bloodstream [37]. As shown in Figure 7D, the release rate of DOX at each pH was faster for the G1PBG$_{29}$-*eg*-PEO/DOX system than for the G2PBG$_{29}$-*eg*-PEO/DOX. For example, the total amount of drug released from G1PBG$_{29}$-*eg*-PEO/DOX after 50 h reached 33.2 and 50.4% at pH 7.4 and at pH 5.5, respectively, as compared with 16.5 and 28.6% for G2PBG$_{29}$-*eg*-PEO/DOX (Figure 7B). This is attributed to the smaller hydrodynamic diameter of G1PBG$_{29}$-*eg*-PEO/DOX relative to G2PBG$_{29}$-*eg*-PEO/DOX, decreasing the diffusion path for drug release and thus increasing the drug release rate. The release profile observed for free DOX is also provided in Figure 7E for comparison. The burst release of DOX is very obvious in that case when compared with all the DOX-loaded micelles.

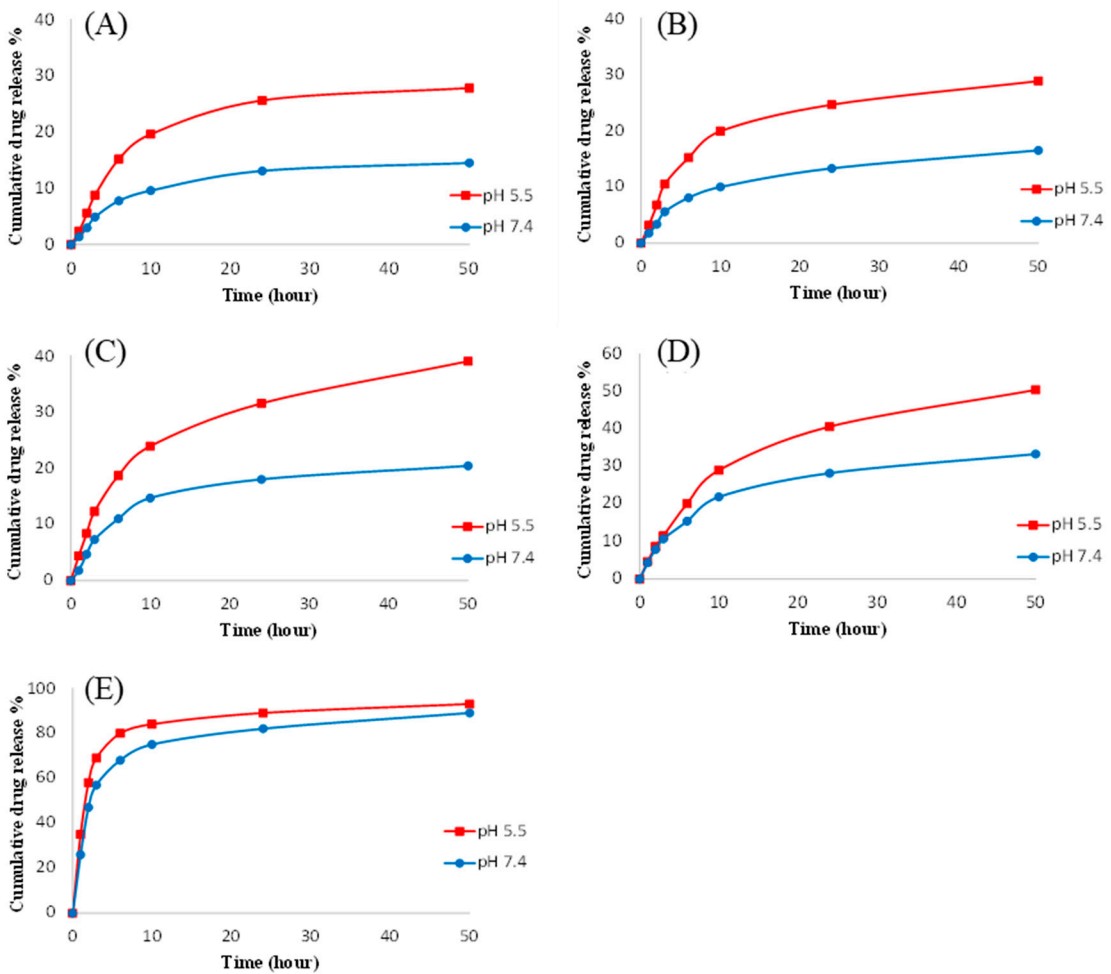

**Figure 7.** In vitro DOX release profiles from (**A**) G2PBG$_{15}$-*eg*-PEO/DOX, (**B**) G2PBG$_{29}$-*eg*-PEO/DOX, (**C**) G2PBG$_{65}$-*eg*-PEO/DOX, and (**D**) G1PBG$_{29}$-*eg*-PEO/DOX unimolecular micelles, and (**E**) free DOX in PBS (pH 7.4 and 5.5) at 37 °C.

As can be seen in Figure 7, faster DOX release was observed from G2PBG$_{65}$-*eg*-PEO/DOX as compared with the other two DOX-loaded micelles, which could be attributed to two factors. The most probable explanation for the difference is the micelle size. Since the drug release in these systems should be diffusion-controlled, faster drug release is expected from micelles with a smaller hydrodynamic diameter due to the shorter diffusion path for drug release. G2PBG$_{65}$-*eg*-PEO/DOX has indeed the smallest hydrodynamic diameter (24 nm) and the fastest DOX release rate as compared with the other two DOX-loaded unimolecular micelles (31 and 27 nm for G2PBG$_{15}$-*eg*-PEO/DOX and G2PBG$_{29}$-*eg*-PEO/DOX, respectively). Another reason could be differences in core branching densities ($b_d$) among the micelles. G2PBG$_{65}$-*eg*-PEO/DOX has the lowest branching density ($b_d$ = 0.11) among the three PBG core substrates and therefore should have a more porous core structure, allowing water molecules to penetrate more easily into it. Consequently, the protonation rate of DOX should increase, resulting in its faster release. In contrast, G2PBG$_{15}$-*eg*-PEO/DOX has the highest branching density ($b_d$ = 0.80) and a more compact core structure (G2PBG$_{15}$), while G2PBG$_{29}$-*eg*-PEO/DOX has an intermediate branching density ($b_d$ = 0.19) which correlates with slower DOX release.

All the DOX-loaded micelles displayed a biphasic DOX release pattern at both pHs, characterized by an initial burst release followed by slower and sustained release of the drug. The burst release could be due to DOX located in the interfacial core–shell region of the micelles, which can diffuse faster into the release medium. In all cases, the release profiles showed incomplete DOX release from the micelles even after one month, as shown in

Figure S7, which is attributed to the dense and branched hydrophobic PBG core interacting strongly with DOX.

All the DOX-loaded chain end-grafted arborescent copolymers exhibited faster DOX release rates in comparison with their randomly grafted counterparts, as shown in Figure S8 [25]. The faster release rates observed in the current investigation are attributed to the smaller size and the unimolecular nature of the end-grafted copolymer micelles. The release rate of DOX was clearly influenced by the pH for both systems, but the chain end-grafted copolymers displayed enhanced pH responsiveness as compared with the randomly grafted systems. Since the chain end-grafted copolymers are expected to have a better-defined core–shell morphology, the hydrophilic PEO chains in the shell should be able to shield the hydrophobic PBG core and the DOX from the aqueous environment more efficiently while minimizing the interfacial region, leading to better-defined pH responsiveness. The results obtained therefore demonstrate the advantages of using chain end-grafted arborescent copolymers rather than randomly grafted copolymers as nanocarriers for drug delivery: The end-grafted copolymers formed either insignificant (G1) or undetectable (G2) amounts of aggregates in aqueous solutions, exhibited more sharply pH-dependent drug release profiles, but also a similar drug loading capacity for DOX, even though their hydrodynamic diameter was smaller than for their randomly grafted analogues.

## 4. Conclusions

In this study, a series of well-defined amphiphilic arborescent copolymers of generations G1 and G2 with three different poly($\gamma$-benzyl L-glutamate) (G0PBG) core branching densities ($b_d$) were synthesized successfully and evaluated as drug delivery nanocarriers. The copolymers were obtained by end-grafting arborescent poly($\gamma$-benzyl L-glutamate) (PBG) cores with different structures of generations G1 and G2, with poly(ethylene oxide) (PEO) chain segments. These copolymers were characterized in solution by DLS, $^1$H NMR, and SEC-LS measurements, as well as by TEM and AFM in the dry state. All the copolymers investigated formed spherical unimolecular micelles with dimensions tailorable from 13 to 29 nm, depending on their generation number and core structure. The encapsulation ability of these unimolecular micelles was evaluated using doxorubicin (DOX) as a model hydrophobic drug, and drug loading contents of up to 11.2 wt% with pH-responsive sustainable drug release were observed by UV-vis spectroscopy.

Both the drug loading content (DLC) and the drug loading efficiency (DLE) increased with the generation number of the copolymers due to the increase in PBG content from G1PBG to G2PBG in the micelles. The branching density of the dendritic G0PBG core was only found to have a minor influence on the overall unimolecular micelle diameter, but it affected the drug loading content, the drug loading efficiency, and the drug release rate more significantly. Increasing the branching density of the G0PBG core by increasing the deprotection level of the substrate led to a larger core diameter and enhanced hydrophobicity for the G2BPG substrate for the synthesis of the unimolecular micelles, which improved their loading capacity for DOX. The release rate of DOX was also affected by the generation number and the hydrophobic core structure for the arborescent copolymers. Increasing the hydrodynamic diameter of the copolymers by increasing the generation number introduced a longer diffusion path for drug release, thus reducing the drug release rate. In addition, the micelles with a denser core structure (G2PBG$_{15}$) exhibited the slowest release rate, while those with a more porous core structure (G2PBG$_{65}$) exhibited the fastest DOX release rate.

The results obtained in the current investigation demonstrate very clearly the advantages of suitably designed arborescent copolymers as unimolecular micelle carriers for drug delivery. Indeed, it was shown that the copolymers containing end-grafted PEO segments were much better than randomly grafted systems [25] at shielding hydrophobic interactions between the hydrophobic PBG cores, thus preventing aggregation of the micelles. Beyond improved dispersibility, the end-grafted copolymer systems displayed increased pH responsiveness relative to the randomly grafted systems. The molecular weight of the hydrophilic PEO chains forming the stabilizing shell was also shown to be an important

parameter in that the end-grafted PEO chains with $M_n$ = 10,000 g/mol PEO chains were much more soluble (and less aggregated) than analogous copolymers constructed with $M_n$ = 5000 g/mol PEO chains [23].

Considering their controllable molecular structure, good water solubility and drug loading capacity, and the desirable pH-dependent controlled release characteristics observed, unimolecular micelles based on amphiphilic arborescent copolymers appear to hold great potential as nanocarriers for drug delivery in cancer treatment.

**Supplementary Materials:** The following supporting information can be downloaded at https://www.mdpi.com/article/10.3390/ijtm3040035/s1, Table S1. Characteristics of linear PBG substrates. Table S2. Characteristics of arborescent G0PBG with different branching densities ($b_d$). Table S3. Characteristics of chain end-functionalized generation G1 and G2 arborescent PBG substrates. Scheme S1. Polymerization of γ-benzyl L-glutamic acid N-carboxyanhydride (Glu-NCA) and deactivation of the terminal amine moiety on the linear PBG substrate, followed by partial or complete deprotection of the PBG substrate. Scheme S2. Polymerization of ethylene oxide with 3-aminopropanol and DPMK. Figure S1. $^1$H NMR spectrum for linear PBG with two *tert*-butyl ester protecting groups at one chain end ($t$BuO)$_2$-PBG. Figure S2. $^1$H NMR spectrum for α-amino PEO in CDCl$_3$. Figure S3. Hydrodynamic diameter distributions of the arborescent copolymers determined by DLS in DMF (left) and in PBS solution (right): (A) G1PBG$_{15}$-*eg*-PEO, (B) G1PBG$_{29}$-*eg*-PEO, and (C) G1PBG$_{65}$-*eg*-PEO. Figure S4. $^1$H NMR spectra for G2PBG$_{15}$-*eg*-PEO in (a) D$_2$O and (b) deuterated DMSO. Figure S5. TEM (left) and AFM phase (right) images for chain end-grafted arborescent copolymers: (A) G1PBG$_{15}$-*eg*-PEO, (B) G1PBG$_{29}$-*eg*-PEO, and (C) G1PBG$_{65}$-*eg*-PEO. Figure S6. $^1$H NMR spectra for (a) DOX in D$_2$O, (b) G2PBG$_{15}$-*eg*-PEO in D$_2$O, and (c) G2PBG$_{15}$-*eg*-PEO/DOX in D$_2$O. The spectra show the appearance of the DOX signals upon encapsulation in G2PBG$_{15}$-*eg*-PEO. Figure S7. In vitro DOX release profiles over 30 days from (a) G2PBG$_{15}$-*eg*-PEO/DOX, (b) G2PBG$_{29}$-*eg*-PEO/DOX, and (c) G2PBG$_{65}$-*eg*-PEO/DOX micelles in PBS (pH 7.4 and 5.5) at 37 °C. Figure S8. In vitro DOX release profiles from (a) G2PBG$_{15}$-*eg*-PEO/DOX and G2PBG$_{15}$-*g*-PEO/DOX, (b) G2PBG$_{29}$-*eg*-PEO/DOX and G2PBG$_{29}$-*g*-PEO/DOX, (c) G2PBG$_{65}$-*eg*-PEO/DOX and G2PBG$_{65}$-*g*-PEO/DOX unimolecular micelles in PBS (pH 7.4 and 5.5) at 37 °C. Reference [38] is listed in the Supplementary Material.

**Author Contributions:** Conceptualization, M.G.; methodology, M.A. and M.G.; validation, M.A. and M.G.; formal analysis, M.A.; investigation, M.A.; resources, M.G.; writing—original draft preparation, M.A.; writing—review and editing, M.G.; supervision, M.G.; project administration, M.G.; funding acquisition, M.G. All authors have read and agreed to the published version of the manuscript.

**Funding:** This research was funded by the Natural Sciences and Engineering Research Council of Canada.

**Institutional Review Board Statement:** Not applicable.

**Informed Consent Statement:** Not applicable.

**Data Availability Statement:** Data can be made available upon request.

**Acknowledgments:** The authors thank Taibah University and the Natural Sciences and Engineering Research Council of Canada (NSERC) for financial support.

**Conflicts of Interest:** The authors declare no conflict of interest.

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
