# Peer review of "Influence of the Core Branching Density on Drug Release from Arborescent Poly(γ-benzyl L-glutamate) End-Grafted with Poly(ethylene oxide)"

_2673-8937, doi:10.3390/ijtm3040035_

Round 1

Reviewer 1 Report

Comments and Suggestions for Authors

The manuscript «Influence of the Core Branching Density on Drug Release from Arborescent Poly(γ-benzyl L-glutamate) End-grafted with Poly(ethylene oxide)» submitted by Mosa Alsehli and Mario Gauthier to Int. J. Transl. Med. is devoted to the synthesis amphiphilic dendritic copolymers and investigation of their properties as nanocarriers for doxorubicin. Authors used Arborescent Poly(γ-benzyl L-glutamate) of generation G1 and G2 End-grafted with Poly(ethylene oxide) (PBG-eg-PEO) as nanocarriers in their work.

The introduction contains the information about design of polymeric micelles and show the advantages of unimolecular structures. The section materials and methods demonstrates description of materials, characterization methods, synthesis of PBG-eg-PEO, their loading by DOX (doxorubicin) and in vitro release of DOX.

Result and discussion section is devoted to the detailed discussion of the synthesis and characterization of new polymers.

I think this work will be interesting for a relatively narrow range of researcher working on the development of new nanocarriers for drug delivery. This research will be good basis for some patent.

I have some questions and remarks:

1)      I think that poly(ethylene oxide) is the same as poly(ethylene glycol). Why author used non-popular in scientific publication name for this compound. I think it will be more comfortable to use PEG in the text.

2)      What molecular mass of PEO (PEG) do authors used in their synthesis?

3)      How PEO-amine was prepared? There are no references or methods in the text about that.

4)      Authors used SEC-MALLS for characterization of copolymers, but there are no description of this method in experimental part or in results. Please add these details.

5)      Abbreviation MALLS is not decoding in the text.

6)      Scheme 1. What are the value for x and y in panel (A)?

7)      Scheme 1. Panel B. You use hydrochloride in the reaction. Why they can react in absence of HCl Neutralization? Please add all conditions of reaction on this scheme.

8)       What is the differences between generation 0, 1 and 2 in your copolymers?

9)      What is the mechanism of interaction of DOX with PBG? What chemical structures are interact. Please add the explanation for scheme 2.

10)  Figure 6. How many times do you repeat the experiment of releasing DOX from complex with copolymers? Do you have any statistically correct results? Please add these data on the figure.

Comments on the Quality of English Language

Minor editing of English language required. Authors must check the manuscript for possible mistaping.

Reviewer 2 Report

Comments and Suggestions for Authors

Dear authors, Dear editor,

In the manuscript entitled “Influence of the Core Branching Density on Drug Release from Arborescent Poly(γ-benzyl L-glutamate) End-grafted with Poly(ethylene oxide” Gauthier and co-worker describe the synthesis and characterization of dendritic amphiphilic polymers that self-assemble into unimolecular micelles. The authors demonstrate that the unimolecular micelles are effective nanocarriers for pH-dependent sustained drug release of doxorubicin in aqueous media. The authors use a robust ensemble of experimental techniques and methodologies for synthesis and characterization of the arborescent polymers and for the characterization of the self-assembled micelles. The authors discuss thoroughly the interplay between the molecular architecture of the polymers and the micelle´s properties. The work is performed to high technical and scientific standards.

I greatly enjoyed reading the manuscript. However, I must express concern regarding the “incremental” nature of this work (?). The authors must clearly highlight in the manuscript the “progress” made here in relation to their previous work (ref 25). The authors should consider to include in this manuscript some “differentiating results” such as cell toxicity/biocompatibility assays?

-        The authors should include in the synthetic or discussion section a scheme showing clearly the architecture of the arborescent polymers to clarify concepts such as generation number, branching density, etc. which are used in the results and discussion section for relating the molecular architecture of the polymers with the micelle´s properties.

-        A scheme of the synthesis, ideally highlighting the advance made in this work in relation to previous works, must be included in the main article. Simply referring readers to previous works is not suitable for a "chemistry paper".

-        Could the authors comment on the reproducibility of the synthesis (e.g., batch to batch reproducibility)?

-        In the drug delivery study, the authors completely remove and replace the release buffer after each time point. In the usual procedure, only a small fraction of the release buffer is removed for analysis. Please discuss the effect of this procedure on the drug release profile.

Best regards

Reviewer 3 Report

Comments and Suggestions for Authors

The authors report a concise work about a drug delivery system using the dendric polymer system with a unique concept. Such drug delivery systems are important for developing various bio medical applications. Such drug delivery systems are developing in the current research interest to overcome the side effect of classical drug administration. Though developing such drug delivery system are not easy task it needs to developed for its potential use. This work holds a good representation about such development. I suggest some major revision on this work to accept the paper.

1.      In the introduction, the authors are very specific about only polymer-based drug delivery development, but drug delivery is a broad concept, I would recommend to introduce comparing other drug delivery system and compare with the reported drug delivery system with more important recent references.

2.      What are the advantages of such dendrimer polymer to be used in drug delivery system because high molecular weight molecules having various limitation in physiological conditions.

3.      Authors reports a details study about synthesis and characterisation but it lacks of biological assays experiments, I would suggest to add some toxicity assay in animal cells to validate the claim of this paper.

4.      Authors characterised only polymers in TEM and AFM, It would better to characterised more specifically after drug loading and released polymers to find the changes with drug delivery system.

5.      When to talked about drug delivery system, it is always better to compare with various drug, rather in simpler way some cation/anionic or hydrophilic/hydrophobic dyes are good candidate to control loading efficiency.

6.      Additional corrections

a. Size profile diagram of the TEM image analysis need to show in figure 4.

b. The hight measurement in AFM images are needed to get idea about the hight of the aggregates.

Reviewer 4 Report

Comments and Suggestions for Authors

The present article entitled “Influence of the Core Branching Density on Drug Release from Arborescent Poly(γ-benzyl L-glutamate) End-grafted with  Poly(ethylene oxide” and authored by Mosa Alsehli and Mario Gauthier is an interesting, comprehensive, and very well written work. The authors present the synthesis of well-defined amphiphilic arborescent copolymers with different poly(γ-benzyl L-glutamate) core branching densities, as well as end-grafted arborescent poly(γ-benzyl L-gluta-mate) cores, with poly(ethylene oxide) chain segments. These polymers were characterized in solution or in dry state by a variety of techniques like DLS, 1H NMR, SEC-MALLS, TEM and AFM, exhibiting the formation of spherical unimolecular micelles with tailorable dimensions, depending on their generation number and core structure. Finally, the copolymers were evaluated as drug delivery nanocarriers. DOX was used as a model hydrophobic drug and the drug release studies were performed considering several parameters like, the pH of the medium, the branching density of the polymers, the generation number and the hydrophobic core structure of the arborescent copolymers, etc.

This work has enough experiments presented, as well as many results and well explained and complete discussion. The conclusions are supported by the experiment results and the references are relative. This work on my part can be accepted in its present form in Int. J. Transl. Med.  So, I have no further comments for the authors.

Round 2

Reviewer 2 Report

Comments and Suggestions for Authors

Dear authors, Dear Editor,

The authors reply fully addressed my concerns regarding the novelty of this work.

The authors addressed also my comments including illustrative schemes in the manuscript.

I believe that the manuscript reached the publication standards.

Best regards.  

Reviewer 3 Report

Comments and Suggestions for Authors

The authors provided a sufficient explanation regarding the reviews, now the paper can be accepted in its present form.